# Endothelial Dysfunction in Chronic Heart Failure: Assessment, Findings, Significance, and Potential Therapeutic Targets

**DOI:** 10.3390/ijms20133198

**Published:** 2019-06-29

**Authors:** Manal M. Alem

**Affiliations:** Department of Pharmacology, College of Clinical Pharmacy, Imam Abdulrahman Bin Faisal University, PO Box 1982, Dammam 31441, Saudi Arabia; malem@iau.edu.sa

**Keywords:** chronic heart failure, endothelial function, endothelial dysfunction, exercise training

## Abstract

Chronic heart failure (CHF) is a complex syndrome that results from structural and functional disturbances that affect the ability of the heart to supply oxygen to tissues. It largely affects and reduces the patient’s quality of life, socio-economic status, and imposes great costs on health care systems worldwide. Endothelial dysfunction (ED) is a newly discovered phenomenon that contributes greatly to the pathophysiology of numerous cardiovascular conditions and commonly co-exists with chronic heart failure. However, the literature lacks clarity as to which heart failure patients might be affected, its significance in CHF patients, and its reversibility with pharmacological and non-pharmacological means. This review will emphasize all these points and summarize them for future researchers interested in vascular pathophysiology in this particular patient population. It will help to direct future studies for better characterization of these two phenomena for the potential discovery of therapeutic targets that might reduce future morbidity and mortality in this “at risk” population.

## 1. Introduction

The endothelium is a single layer of squamous epithelial cells called endothelial cells, which line the inside of all blood vessels. In a healthy state, the endothelium is a dynamic tissue whose constituent cells synthesize and secrete substances affecting its own function and that of adjacent structures. It is essentially involved in the regulation of blood flow, blood coagulation, vascular permeability, and structure. Endothelial dysfunction is a pathological condition characterized by loss of balance in all major endothelial mechanisms. It has been implicated in the pathophysiology of different cardiovascular diseases, including chronic heart failure [1]. Its existence in CHF patients imposes increased morbidity and mortality, and if established, it might persist even after cardiac transplantation [2]. Despite that, it is still debatable whether it underlies all types of chronic heart failure (CHF): Ischemic vs. non-ischemic, systolic vs. diastolic, and mild vs. severe [3]. Therefore, endothelial dysfunction in CHF needs well-structured research for a better understanding of how it develops, in which patients’ groups, its significance, and its reversibility by pharmacological and non-pharmacological interventions. This review will summarize the literature for future investigators pursuing that field of research. It used scholarly platforms and several libraries, including PubMed, Google Scholar, and Science Direct, to extract data on endothelial function/dysfunction in chronic heart failure.

## 2. Assessment of Endothelial Function

The principle of endothelial function assessment involves measuring the vasodilatory response to endothelium-dependent agonists, such as acetylcholine (Ach), carbachol, methacholine, serotonin (5-hydroxytryptamine, 5-HT), and others, as all induce endothelial cells (ECs) to generate nitric oxide (NO), and then comparing this response to that of agonists that are endothelium-independent, such as sodium nitroprusside (SNP) or glyceryl trinitrate (GTN). These vasoactive substances can be infused locally into forearm blood vessels via a technique called venous occlusion plethysmography (VOP).

VOP is a technique that was first described over 100 years ago and is still considered as a “gold-standard” investigation for the assessment of vascular physiology *in vivo* in different populations. It is commonly applied to the forearm vascular bed although it can also be applied to lower limbs. The aforementioned vasoactive substances can be infused intra-arterially into the brachial artery under local anesthesia to measure forearm blood flow (FBF) in mL/min/100 mL [4]. Changes in forearm blood flow result in changes in arm circumference and a change in the strain gauge length that is placed around the forearm. The other non-infused arm will provide the control blood flow. Alternatively, blood flow can be measured by Doppler flow measurements in the test arm that can be related to the control arm as a percentage.

Due to the invasive nature of VOP, flow-mediated dilatation (FMD) was introduced as a non-invasive technique to measure flow-mediated changes in arterial diameters in relatively superficial arteries, such as brachial, radial, or femoral arteries [5]. In principle, endothelium-dependent FMD can be assessed by an ultrasound system equipped with two-dimensional (2D) imaging, color, and spectral Doppler. A baseline image of the brachial artery is obtained, followed by arterial occlusion by blood pressure cuff inflation to supra-systolic pressure to occlude arterial inflow, which causes ischemia. Subsequent cuff deflation induces a high-flow state (reactive hyperemia) in the brachial artery, with shear stress causing vasodilatation that can be measured [6]. The change in brachial artery diameter can be quantified as a percentage change from the baseline diameter.

## 3. Findings

The first description of endothelial dysfunction was in 18 patients with essential hypertension who were compared with 18 control subjects in 1990 [7]. Such a phenomenon was confirmed later with other studies and attributed to systemic oxidative stress and vascular inflammation [8]. As another important risk factor for heart failure, type 2 diabetes mellitus was also associated with endothelial dysfunction. Such pathology was attributed to systemic oxidative stress due to hyperglycemia, fatty acid excess, and insulin resistance [9,10]. ED at the coronary microvascular level in diabetic patients could precipitate coronary microvascular dysfunction and ischemic heart disease in the absence of coronary artery disease [11].

The description of endothelial dysfunction in the heart failure population started to appear in the literature and was mainly performed on a small scale. A study that included 24 patients with CHF with left ventricle (LV) systolic dysfunction (New York Heart Association NYHA class II–III) demonstrated that endothelium-dependent vasodilatation was impaired in patients when compared with 22 control subjects. Such a finding was demonstrated invasively via the VOP technique with methacholine infusion into the brachial artery with FBF of 5.32 ± 0.31 in CHF patients vs. 9.52 ± 0.60 mL/min/100 mL in control subjects; *p* = 0.0003. FBF in response to nitroprusside infusion (endothelium-independent vasodilatation) was not significantly different between the two groups [12].

Endothelial dysfunction was assessed in nine patients with CHF with LV systolic dysfunction (NYHA class III) and in nine control subjects invasively via the VOP technique with acetylcholine and NG-monomethyl-L-arginine (L-NMMA) as NO synthase inhibitor infusion in the brachial artery. FBF in response to acetylcholine was blunted in CHF patients (endothelium-dependent vasodilatation). In addition, there was an exaggerated decrease in blood flow induced by L-NMMA infusion (nitric oxide derived from L-arginine is preserved/enhanced) in CHF patients when compared with control subjects [13].

Conduit artery distensibility is determined, at least, in part, by endothelial function. Such a method was used to assess endothelial function in nine patients with CHF with LV systolic dysfunction (NYHA class I–III) and in nine control subjects. There was a reduction in the pulse wave velocity in the right common iliac artery in response to acetylcholine local infusion in control subjects but not in CHF patients. The same study assessed the brachial artery diameter and distensibility change in response to reactive hyperemia in the hand. Such a technique has shown increased parameters in control subjects (8.8% and 18.4%, respectively) but not CHF patients (0.3% and −4.5%) for similar blood flow and similar GTN induced effects [14].

A fourth study demonstrated the existence of endothelial dysfunction in 14 patients with CHF with LV systolic dysfunction (NYHA class III) who were compared with 10 control subjects. Flow-dependent endothelium-mediated vasodilatation (FDD) assessed the radial artery diameter change in response to wrist occlusion. FDD was impaired in patients with CHF in comparison with control subjects (*p* < 0.01). Such a phenomenon was interestingly associated with increased endothelium-bound xanthine oxidase activity and reduced endothelium-bound extracellular superoxide dismutase activity [15].

Later studies started to include patients having heart failure with preserved ejection fraction (HFpEF) as the target population. Table 1 lists four studies, where the first study assessed endothelial function by brachial artery-FMD (BA-FMD) in comparison with patients with systemic hypertension. The second and third studies assessed BA-FMD, along with an assessment of the microvascular endothelial dysfunction in comparison with hypertensive and healthy controls, respectively. The fourth assessed BA-FMD along with an intima-media thickness measurement as an index of an abnormal vascular structure. Collectively, these studies show the existence of endothelial dysfunction in HFpEF patients in conduit arteries and the microvascular circulation.

The suggested pathophysiological mechanism of ED in CHF is related to a state of increased oxidative stress in this patient population via multiple mechanisms, reduced NO synthesis [20] with possible involvement of genetic polymorphism for endothelial nitric oxide synthase (eNOS) [21], oxidative inactivation of NO [22], increased levels of asymmetric dimethylarginine (ADMA), an endogenous eNOS inhibitor [23], increased plasma oxidized low- density lipoprotein concentrations [24], increased 8,12-isoprostane F (2α) concentration [25], increased endothelium-bound xanthine oxidase (XO) activity, and reduced endothelium-bound extracellular superoxide dismutase activity [15]. The latest research revealed the role of microRNAs (miRs) in endothelial (dys) function [26], and demonstrated that circulating miR-126 released from ECs is downregulated in CHF [27]. Such a property might have implications in endothelial dysfunction in this patient population.

## 4. Significance

### 4.1. Predictive Value of ED in Population-Based Studies

Endothelial dysfunction assessed by brachial artery FMD was a predictor of 5 years incident cardiovascular events in a case-cohort sample of 3026 subjects with traditional cardiovascular risk factors that participated in the Multi-Ethnic Study of Atherosclerosis. FMD findings improved the classification of those subjects as low, intermediate, and high CVD risk compared with the Framingham Risk Score (FRS) [28]. Similar findings were reported in two other smaller cohort studies. The first included 921 subjects with traditional cardiovascular risk factors who participated in the Prospective Study of the Vasculature in Uppsala Seniors (PIVUS) study, where endothelial dysfunction was assessed invasively by the VOP and independently associated with 5-year risk of a composite end point of death, myocardial infarction, or stroke [29]. The second cohort study included 435 subjects with traditional cardiovascular risk factors but no apparent heart disease, where FMD of the brachial artery independently predicted long-term adverse cardiovascular events after a follow-up period of 32 months, in addition to other risk factors’ assessment [30].

Two other studies were less supportive of the predictive value of endothelial dysfunction when compared with another measure of vascular structure, such as the intima-media thickness. The first included 444 patients described as being at risk of cardiovascular events because of underlying coronary artery disease (CAD), renal dysfunction, or the presence of multiple cardiovascular risk factors, who were followed for 24 months. This study has shown that patients with the most severe endothelial dysfunction assessed by brachial artery FMD had greater subsequent cardiac morbidity and mortality compared with those with normal or mild endothelial dysfunction. However, mortality in such a population was independently predicted by carotid intima-media thickness (IMT) and left ventricular mass rather than FMD [31]. The latter study included 398 patients for whom coronary angiography was performed due to chest pain, along with brachial artery FMD and brachial artery IMT. After a follow-up period of 39 months, only the presence of CAD and brachial artery IMT significantly predicted future cardiovascular events [32].

### 4.2. Predictive Value of ED in the CHF Population

Table 2 summarizes seven studies that assessed endothelial function in CHF patients. All of these studies were performed on patients with heart failure with systolic dysfunction (i.e., heart failure with reduced ejection fraction HFrEF). They included patients of all severity classes, with ischemic and non-ischemic etiologies. Endothelial dysfunction was assessed at a medium-sized conduit, and small arteries were found to be an independent predictor of mortality and hospitalization due to worsening of HF during a follow-up period that varied between 1 and 5 years.

Two other studies assessed the significance of endothelial function in CHF after they had undergone major interventions. The first included 185 CHF patients (of ischemic and non-ischemic etiology), 25 months after they received heart transplantation (EF of 75 ± 10%; SD). Epicardial endothelial function, assessed in response to intracoronary acetylcholine administered during coronary angiography, was found to be an independent predictor of 5-year cardiovascular-related events and death after heart transplantation [40]. The second study included 34 patients with CHF with systolic dysfunction (of ischemic and non-ischemic etiology) who were treated with cardiac resynchronization therapy for the treatment of advanced HF. Their baseline peripheral endothelial function (RH-PAT) independently predicted hospitalization due to HF progression during a follow-up period of 11 months [41].

## 5. Potential Therapeutic Targets for ED in the CHF Population

### 5.1. Role of Exercise Training on ED in HFrEF

In the general population, physical activity is an essential health strategy that needs to be encouraged. Based on the results of a large cohort study that included more than 20,000 middle- aged men and women, regular physical activity (bicycling to work) was related to a lower risk of incident obesity, hypertension, hypertriglyceridemia, and impaired glucose tolerance in comparison with passive travel during 10 years of follow-up [42]. For patients with HF, the European Society of Cardiology ESC 2016 class IA level recommendations encourage regular aerobic exercise to improve functional capacity and symptoms. This recommendation is regardless of the LV ejection fraction (LVEF). Regular aerobic exercises are also encouraged in stable patients with HFrEF to reduce the risk of HF hospitalization [43]. Since endothelial dysfunction plays an important role in the physical incapacity of HF patients, studies have shown that factors, such as reduced NO bioavailability, increased oxidative stress, increased pro-inflammatory cytokines, reduced number of circulating endothelial progenitor cells, and reduced functional capacity of circulating angiogenic cells, all underlie endothelial dysfunction in this patient population. Hence, exercise training has shown its genuine ability to combat these contributing factors, at least, partially and to improve endothelial function accordingly [44]. For that reason, numerous studies in the literature have shown the advantage of exercise training on endothelial function in heart failure patients (HFrEF as it is the most abundant patient group studied so far). Such a benefit was quantified with a recent meta-analysis that included a total of 529 individuals (NYHA class I–III) who participated in aerobic, resistance, and functional electrical stimulation (FES) for an intervention duration that ranged between 4 weeks and 6 months. Such an analysis has shown that overall exercise training improved FMD with a standardized mean difference (SMD) of 1.08 (95%CI 0.70–1.46, *p* < 0.001) [45]. A later smaller study that included 35 patients with more advanced CHF (NYHA class III–IV) was published; it showed a trend of a similar benefit in FMD with low-frequency electrical muscle stimulation for 8 weeks when compared with skin only stimulation (SHAM) [46]. Suggested mechanisms of exercise training benefits were increased NO bioavailability by shear stress [47], upregulation of eNOS expression and phosphorylation [48], increased antioxidant enzymes [49], and lately, mobilization of bone marrow-derived endothelial progenitor cells (EPCs) that are involved in the endogenous repair of endothelial dysfunction [50,51].


**Adrenergic System, Physical Exercise, and ED in HFrEF Patients**
The sympathetic nervous system’s (SNS) over activity in heart failure is a well-established mechanism that has a transitional contribution to the pathophysiology, is supportive of cardiac function initially, and is a burden on it ultimately [52]. Research has revealed molecular alterations occurring in SNS in different models, myocardial cells, peripheral lymphocytes, and endothelial cells. Interpretation of these changes should take into consideration genetic polymorphism in human adrenergic receptors (ARs) and NOS enzymes, which adds to the complexity of the pathophysiological interactions of these phenomena in heart failure.In the myocardium, the most abundant AR are βAR subtypes. Their density changes from a predominant β_1_AR (77%): β_2_AR (23%) in a non-failing heart to 60%: 38% with improper signaling mechanisms in a failing heart [53,54]. βAR desensitization has also been reported in the myocardium of heart failure patients [55,56], a property that has also been associated with receptor phosphorylation by adrenergic receptor kinase-2 [57]. Peripheral blood lymphocytes served as another model to study changes in βAR in heart failure patients. This is because they exist in high numbers in T_suppressor/cytotoxic_ cells, B cells, natural killer cells, and monocytes [58]. These numbers have been found to correlate with βAR density in human myocardial cells [59]. In heart failure patients, the density of βAR on peripheral lymphocytes and isoproterenol-stimulated adenylate cyclase activity were reduced [60].The benefits of physical exercise in heart failure were studied on myocardial cells and peripheral lymphocytes. In animal models, physical exercise increased myocardial βAR density and reduced G-protein-coupled receptor kinase-2 over-expression, with the resultant improvement of cardiac inotropic activity in response to adrenergic stimulation [61]. In patients with heart failure, physical exercise has increased the density of βAR in peripheral lymphocytes [60]. The third studied model are endothelial cells (ECs), where all AR subtypes exist (with β_2_AR sub-type predominance) and directly or via NOS stimulation induce NO-dependent vasodilatation [62,63]. α_2_AR are also expressed and involved in the activation of NOS and NO-dependent vasodilatation [64].The benefits of physical exercise in heart failure were studied on myocardial cells and peripheral lymphocytes. In animal models, physical exercise increased myocardial βAR density and reduced G-protein-coupled receptor kinase-2 over-expression, with the resultant improvement of cardiac inotropic activity in response to adrenergic stimulation [61]. In patients with heart failure, physical exercise has increased the density of βAR in peripheral lymphocytes [60]. The third studied model are endothelial cells (ECs), where all AR subtypes exist (with β_2_AR sub-type predominance) and directly or via NOS stimulation induce NO-dependent vasodilatation [62,63]. α_2_AR are also expressed and involved in the activation of NOS and NO-dependent vasodilatation [64].Considering the current findings, we could anticipate that physical exercise in heart failure might modify the molecular changes to the adrenergic system at the EC level and improve endothelial function, as it does enhance autonomic receptor-mediated endothelium-dependent vasodilatation in animal models [65]. Such a benefit might be in addition to its ability to recruit EPCs from the bone marrow to repair endothelial function. Therefore, future studies should establish which exercise regimens should be emphasized and to which heart failure patients. In addition, more research is needed to establish the effect of physical exercise on prognosis if continued or terminated.

### 5.2. Role of Exercise Training on ED in HFpEF

There is a paucity of data in the literature on HFpEF patients. A study included 30 patients with HFpEF (NYHA class II–III) who were randomized to functional electrical stimulation (FES) (25 Hz; 15 patients) vs. a placebo “much lower intensity of stimulation” (5 Hz; 15 patients) for 6 weeks. The FES group had a significant improvement in BA-FMD, 6-min walking distance, and quality of life with the HF questionnaire compared with the placebo group [66]. However, another study, which included 63 patients with HFpEF (NYHA class II–III) who were randomized to endurance training (32 patients to walking, arm, and leg ergometry, and 31 to attention control) for 16 weeks, did not show any improvement in endothelial function assessed by the same technique [67].

### 5.3. Role of Pharmacological Agents on ED in HFrEF

Pharmacological agents explored in the literature include those considered standards of care in the HF population and other newly investigated agents. Standard heart failure therapies include angiotensin-converting enzyme (ACI) inhibitors, angiotensin receptor blockers (ARBs), spironolactone, and the commonly co-administered statins. ACE inhibitors’ beneficial effect on endothelial function was described in the literature, mostly in the hypertensive population, with a meta-analysis that demonstrated that they significantly improved BA-FMD when compared with placebo, no treatment, and other antihypertensive agents, like calcium channel blockers and beta-blockers [68]. Further experimentation of their potential additive benefit if combined with ARBs in the hypertensive population was carried out in two studies with inconsistent results [69,70]. Spironolactone’s short-term administration has shown beneficial effects on endothelial function assessed by brachial artery VOP, which was associated with increased vasoconstriction with L-NMMA infusion in 10 patients with CHF with LV systolic dysfunction with ischemic etiology (NYHA class II–III) [71]. In another small study that included 20 patients with CHF with LV systolic dysfunction with ischemic etiology (NYHA class III–IV), it showed a beneficial effect on BA-FMD as compared with baseline readings [72]. Co-administered lipid-lowering therapy in HF patients is commonly achieved with statins, which might have an additional benefit on endothelial function in such a population. A 12-week administration of rosuvastatin in high doses (40 mg/d) improved endothelial function assessed by radial artery-FMD in 42 patients with CHF with LV systolic dysfunction with ischemic and non-ischemic etiology (NYHA class II–III) as compared with the placebo [73]. A smaller dose of Rosuvastatin (10 mg/d) administered over 1 month to 21 patients with similar criteria also resulted in a significant improvement in BA-FMD and released EPCs as compared with baseline figures [74]. In a third study, atorvastatin in two doses (10 and 40 mg/d) administered over 4 weeks in 26 patients with CHF with LV systolic dysfunction with ischemic etiology (NYHA class II–III) resulted in a significant improvement in BA-FMD and the number of bone-marrow release of EPCs in a dose-dependent manner [75]. These positive findings were similar to those documented with atorvastatin 20 mg daily dose administered over 8 weeks in 38 CHF with LV systolic dysfunction with ischemic and non-ischemic etiology (NYHA class II–IV) [76].

In the current literature, new pharmacological agents have started to appear to improve endothelial function in different patient populations. Among them, allopurinol as a XO inhibitor showed some benefit to endothelial function that is most noticeable in CHF patients. This is because circulating XO can bind to the endothelial surface where reactive oxygen species (ROS) are produced to contribute to endothelial dysfunction [77], and because uric acid per se has been found to correlate negatively with FMD readings [78]. In a recent meta-analysis that included a total of 197 patients with CHF with LV systolic dysfunction with ischemic and non-ischemic etiology (NYHA class II–III), allopurinol 300 mg dose administered for a period of 1 week to 3 months improved endothelial function with a standardized mean difference of 0.776 (95%CI; 0.429–1.122, *p* < 0.001) [79].

Levosimendan is a calcium-sensitizing agent used for the management of HFrEF. It has been shown to possess vasodilatory properties via inhibition of phosphodiesterase III, desensitization of contractile proteins to calcium, and by the opening of ATP-sensitive K^+^ channels. In addition, it induces eNOS-dependent NO production in coronary endothelial cells in animal models [80]. It has also been shown to improve endothelial function assessed via FMD in 26 patients with CHF with LV systolic dysfunction with ischemic and non-ischemic etiology (NYHA class II–III). Such a benefit was associated with the inhibition of vascular inflammation [81].

After the beneficial effects of ivabradine on clinical outcomes in CHF patients with LV systolic dysfunction [82] (SHIFT trial), and its benefit on ventricular-arterial coupling (VAC), its effect on endothelial function and VAC was assessed in 30 CHF patients with LV systolic dysfunction due to hypertensive heart disease predominantly (NYHA class II). Ivabradine was administered with a dose of 10 mg/d for 4 months, and it showed an improvement in endothelial function at the microvascular level [83].

### 5.4. Role of Pharmacological Agents on ED in HFpEF

Since the optimal therapy of patients with HFpEF is still under investigation, the current literature lacks adequate studies on pharmacological therapies for ED in this patient population. A small study included 48 patients with HFpEF and assessed the effect of sildenafil, a phosphodiesterase 5 inhibitor, administered over 24 weeks, with no significant benefits on the reactive hyperemic change in digital blood flow [84]. With the first large animal model of hypertensive heart disease with a loss of LV capacitance similar to patients with HFpEF, identification of the titin isoform shift and NOS uncoupling in the presence of preserved ejection fraction will help with gaining a better understanding of the pathophysiology and represents future therapeutic targets for pharmacological intervention in this patient population [85].

## 6. Conclusions

Chronic heart failure is a complex syndrome with significant morbidity and mortality worldwide. New staging systems have emerged to address the involvement of not only the heart but also lungs, kidneys, liver, and central nervous system [86]. Endothelial dysfunction is an important concomitant characteristic that plays an integral role in the pathological process and the multi-system involvement. It is systemic in nature and involves arteries and veins. Peripheral endothelial dysfunction assessed via VOP or FMD was found to be a significant and independent predictor of mortality, cardiovascular events, and hospitalization due to worsening HF. This finding seems to be irrespective of severity or etiology of CHF. Additionally, one of the important intervention strategies in the literature to combat this phenomenon is physical exercise. Physical exercise has shown recognized benefits on endothelial function, oxidative stress, skeletal muscles, hemodynamics, and the adrenergic system in patients with HFrEF (Figure 1). In HFrEF patients, the recommended exercise regimen and to which patients is still to be clarified. As exercise training in this population has started to show diversity in the designed regimens, a spectrum that ended in stretching exercises to CHF patients with implantable cardioverter defibrillators has documented success [87]. However, and according to the current literature, less clear benefits of physical exercise are shown in HFpEF [88] Another important consideration for future research is to focus on endothelial repair mechanisms in CHF patients by EPCs mobilized from the bone marrow to facilitate a better understanding of their epigenetic mechanisms that play a role in the repair process, such as DNA methylation, post-translational histone modification, and the expression patterns of some microRNAs (miRNAs) [51]. As far as pharmacological agents are concerned, the strongest evidence available is supportive of allopurinol as a XO inhibitor. Such a drug possesses antioxidant properties and has shown promising benefits on endothelial function in patients with HFrEF. Therefore, allopurinol is worthy of further testing in well-designed clinical trials to explore the value of including it as an add-on therapy to improve clinical outcomes in this “at risk” population.

## Figures and Tables

**Figure 1 ijms-20-03198-f001:**
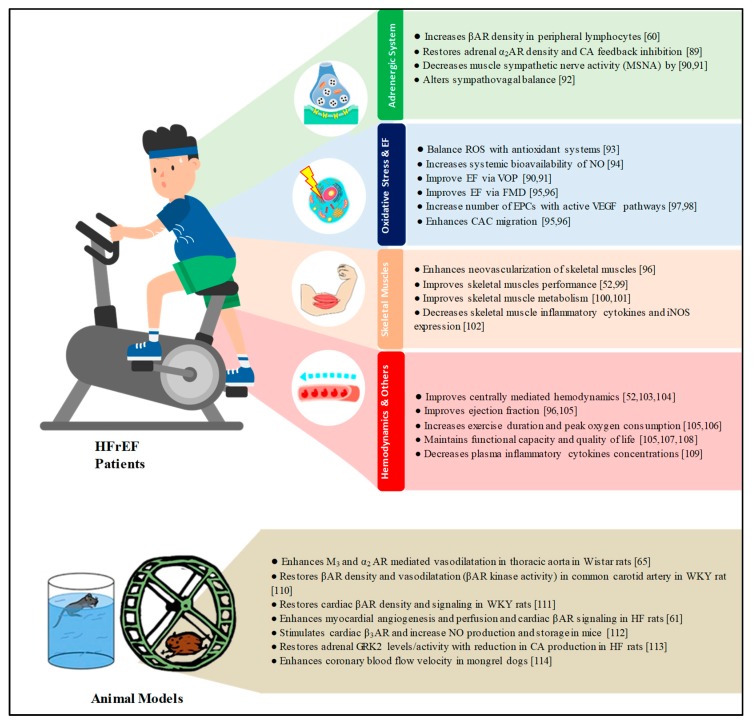
Mechanisms of beneficial effects of physical exercise in HFrEF patients and animal models [89,90,91,92,93,94,95,96,97,98,99,100,101,102,103,104,105,106,107,108,109,110,111,112,113,114]. AR; adrenergic receptors, CA; catecholamines, ROS; reactive oxygen species, NO; nitric oxide, EF; endothelial function, VOP; venous occlusion plethysmography, FMD; flow-mediated dilatation, EPCs; endothelial progenitor cells, VEGF; vascular endothelial growth factor, CAC; circulating angiogenic cells; WKY rats; Wistar–Kyoto rats, HF; heart failure, GRK2; G protein-coupled receptor kinase-2.

**Table 1 ijms-20-03198-t001:** Studies that assessed endothelial dysfunction in patients with heart failure with preserved ejection fraction (HFpEF).

Study	Sample Size and Population	Methodology	Findings
Farrero et al. 2014 [16]	28 HFpEF	Brachial artery FMD	Patients 1.95 (−0.81–4.92)% (median, IQR) Controls 5.02 (3.90–10.12)% *p* = 0.002
	42 Hypertensive control subjects		
Maréchaux et al. 2016 [17]	45 HFpEF	Brachial artery FMD	Patients 3.6 (0.4–7.4)% (median, IQR) Controls 7.2 (3.2–17.2)% *p* = 0.001
	45 Hypertensive control subjects	Laser Doppler Flowmetry to assess forearm cutaneous peak blood flow	Patients 135 (104–206) PU Controls 177 (139–216) PU *p* = 0.03
Lee et al. 2016 [18]	24 HFpEF	Brachial artery FMD	Patients 3.06 ± 0.68% (mean ± SEM) Controls 5.06 ± 0.53% *p* = 0.03, with no difference when corrected to shear rate
	24 healthy controls	Microvascular function via reactive hyperemia (RH)	Patients 454 ± 35 mL/min (mean AUC ± SEM) Controls 659 ± 63 mL/min *p* = 0.03
Kishimoto et al. 2017 [19]	41 HFpEF	Brachial artery FMD	Patients 2.9 ± 2.1% (mean ± SD) Controls 4.6 ± 2.7% *p* = 0.0002
	165 control subjects with cardiovascular risk factors	Brachial artery IMT	Patients 0.35 ± 0.06 mm Controls 0.31 ± 0.07 mm *p* = 0.0002

PU; Perfusion unit, IMT; intima-media thickness.

**Table 2 ijms-20-03198-t002:** Predictive value of endothelial dysfunction in HFrEF patients.

Study	Sample Size and Population	Methodology	Follow-up	Outcome	Findings
Meyer et al. 2005 [33]	75 CHF with systolic dysfunction NYHA class I–IV Ischemic and non-ischemic etiology	BA-FMD Response to reactive hyperemia	3 years	Conversion to United Network of Organ Sharing UNOS status 1 (chronic inotropic support or implantation of ventricular assist device) or death	27 UNOS-1/death 48 survivors
	2 control groups (19 healthy and young subjects, and 14 age- and gender- matched control subjects)				Multivariate stepwise analysis showed that FMD (Chi-square = 11.5, *p* = 0.0007), log BNP (Chi-square = 8.7, *p* = 0.003), and mean BP (Chi-square = 3.9, *p* = 0.047) were independent predictors of the combined endpoint
Heitzer et al. 2005 [34]	287 CHF with mild systolic dysfunction NYHA class I Ischemic and non-ischemic etiology	VOP Response to Ach and SNP	4.8 years	Death, heart transplant, readmission due to worsening HF	79 patients had events 208 patients without
					Cox-proportional hazards model showed that age (HR = 1.07, 95%CI; 1.03–1.11, *p* = 0.001), renal function (HR = 0.97, 95%CI; 0.94–1.02, *p* = 0.001), and blunted Ach-induced vasodilatation (HR = 0.96, 95%CI; 0.94–0.98, *p* = 0.007) were independent predictors of the outcome
Fischer et al. 2005 [35]	67 CHF (30 had systolic dysfunction) NYHA class II–III Ischemic and non-ischemic etiology	FDD Radial artery	3.8 years	Cardiac death, hospitalization due to worsening HF, or heart transplant	24 patients had events 43 patients without
					Cox-regression analysis showed that FDD (HR = 0.665 ± SE 0.182, *p* < 0.01), DM (HR = 0.055 ± 0.946, *p* < 0.01), and EF (HR = 0.054 ± 0.894, *p* < 0.01) were independent predictors of the outcome
Katz et al. 2005 [36]	259 CHF with systolic dysfunction NYHA class II–III Ischemic and non-ischemic etiology	BA-FMD Response to reactive hyperemia in 149 patients	2.3 years	Death, and urgent transplantation	17 patients had events 132 patients without
		Exhaled NO production during submaximal exercise (pulmonary circulation) in 110 patients	1 year		19 patients had events 91 patients without Cox- multivariate proportional- hazards model showed that FMD (HR for 1% decrease in FMD = 1.20; 95%CI; 1.03–1.45; *p* = 0.027), and exhaled NO (HR for 1-ppb/min decrease = 1.31, 95%CI; 1.01–1.69, *p* = 0.04) were independent predictors of the outcome
Shechter et al. 2009 [37]	82 CHF with systolic dysfunction NYHA class IV (advanced) Ischemic etiology	BA-FMD Response to reactive hyperemia	1.2 years	Death, hospitalization for CHF exacerbation, or MI	30 patients had events 52 patients without
					Cox-proportional hazard model showed that FMD (HR for 1% decrease = 1.20; 95%CI; 1.01–1.69; *p* < 0.03) was an independent predictor of mortality
de Berrazueta et al. 2010 [38]	242 CHF with systolic dysfunction NYHA class I–IV Ischemic and non-ischemic etiology	VOP Response to reactive hyperemia	5 years	Total events (death, heart attack, angina, stroke, NYHA class IV, or hospitalization for worsening HF)	737 total events 62 patients died 180 patients survived
					Cox-regression hazard model showed that FBF post-hyperemia (HR = 0.665 ± SE 0.182, *p* = 0.01) was an independent predictor of total events
Fujisue et al. 2015 [39]	362 HFrEF NYHA class I–III Ischemic and non-ischemic etiology	RH-PAT Reactive Hyperemia-Peripheral Arterial Tonometry (peripheral microvascular EF; distal finger)	3 years	HF-related events (composite of cardiovascular death and HF hospitalization)	82 patients had events 280 patients without
					Cox-regression hazard model showed that Ln-RH-PAT (per 0.1, HR = 0.84, 95%CI; 0.75–0.95, *p* = 0.005); Serum sodium (per meq/L, HR = 0.92, 95%CI; 0.87–0.98, *p* = 0.004); and Ln-BNP (per 1.0, HR = 1.38, 95%CI; 1.12–1.70, *p* = 0.002) were independent predictors of HF-related events

BA-FMD; brachial artery flow- mediated dilatation, BNP; B-type natriuretic peptide, BP; blood pressure, VOP; venous occlusion plethysmography of brachial artery, HF; heart failure, Ach; Acetylcholine, SNP; sodium nitroprusside, HR; Hazard Ratio, FDD; flow-dependent, endothelium-mediated vasodilatation, DM; diabetes mellitus, EF; ejection fraction, ppb; parts per billion, MI; myocardial infarction, NYHA; New York Heart Association, FBF; forearm blood flow, EF; endothelial function.

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
