# Peer review of "Endothelial Dysfunction in Chronic Heart Failure: Assessment, Findings, Significance, and Potential Therapeutic Targets"

_ijms, 2019, doi:10.3390/ijms20133198_

Round 1
Reviewer 1 Report
I read with great interest the review manuscript by Manal M. Alem on endothelial dysfunction in chronic heart failure. I think the objective of the review is appealing. I only have a few revisions:
1. I would ask the Author to elucidate which is the reason to build the review on “chronic” heart failure and not on heart failure.
2. Please correct the typing errors made in using“ ; ”instead of “ : ”.For example in the sentences “ including; PubMed, Google scholar” or in “such as; Acetylcholine (Ach), carbachol “ or in “ technique called; venous “ and so on.
3. Please correct the typo in the sentence “afro-mentioned vasoactive”.
4. Please use NYHA class, not Functional class in the sentence “CHF with LV systolic dysfunction (Functional class III)”.
5. In my opinion, HF is a complex syndrome that lead systemic disease with involvment of lung, kidney, liver and so on, in which ED may represent an important factor to be evaluated. In fact, ED can be a common element with pulmonary, renal, hepatic dysfunction that often coexist with HF. [J Am Coll Cardiol. 2014 May 20;63(19):1959-60]
6. I would suggest to add a section on the role of ED in diabetes mellitus that represents one of the more important HF risk factors. [Int J Mol Sci. 2018 Mar 10;19(3). pii: E802. Int J Mol Sci. 2019 Apr 3;20(7). pii: E1658]
7. I would ask the Author to mention the role played by genetic susceptibility in ED: for example polymorphisms for eNOS lead to ED and, then, myocardial ischemia, that is one of the most important etiology for HF. [Basic Res Cardiol. 2013 Nov;108(6):387]
8. In the pharmacological section, I would suggest to add also Levosimendan which down-regulates the expression of the pro-thrombotic and anti-fibrinolytic biomolecules and induces NO production [Atherosclerosis. 2008 Mar;197(1):278-82. Br J Pharmacol. 2009 Jan;156(2):250-61.]
Author Response
Thank you very much for your constructive comments.
1. The reason for building the review on “chronic heart failure” not heart failure;
a) The latest 2016 ESC guidelines for the diagnosis and treatment of acute and chronic heart failure still describes chronic heart failure (CHF) as an entity where a patient has heart failure (HF) for some time, as a terminology related to the time course of heart failure. (European Heart Journal, Volume 37, Issue 27, 14 July 2016, Pages 2129–2200), with acute decompensated heart failure as another entity.
b) Other terminologies related to LV ejection fraction, as well as terminologies related to severity of symptoms (NYHA functional class) are also used in the review
c) The description of patients population in the individual studies included in this review was given as chronic heart failure, and clinical stability was described for approximately 3 months. Accordingly I have quoted theirs.
d) Endothelial function assessment in the literature was mostly done on chronic heart failure patients. Accordingly I wanted to avoid generalization of the title of this review.
2. Typo errors have been corrected
3. Typo errors have been corrected
4. Correction was done
5. The conclusion section has been modified accordingly
6. A small paragraph has been added in findings section before discussing ED in CHF patients; that is ED in systemic hypertension and diabetes mellitus as two important risk factors for HF.
7. Genetic polymorphism for endothelial nitric oxide synthase eNOS was added to the suggested pathophysiological mechanism of ED in CHF.
8. A new paragraph was included in the pharmacological section about levosimendan and the two studies were referred to.

Reviewer 2 Report
I agree with the author that a review on endothelial dysfunction (ED) and chronic hart failure (CHF) would be helpful to the field. I believe this review is a solid effort toward compiling the existing studies on ED in CHF. I do believe that there are areas for improvement in the presentation of this review. Specifically,
1. Overall English editing to conform to standard language and correcting errors. It is readable, but there are a number of errors and incorrect usage and grammar.
2. Introduction, paragraph 1: "The endothelium is a single layer of squamous epithelial cells...." Should that be endothelial cells?
3. Introduction, paragraph 3: "Afro-mentioned" should be "afore-mentioned."
4. Table 1, "Findings": What do the percentages mean? For example is Patients 1.95%/Controls 5.02% supposed to be % risk of developing dysfunction? % physiological function? Something else?
5. Table 2 is cut off on p. 5. Also, since Table 1 is about patients with HFpEF, and Table 2 is about patients with HFrEF, Table 2 should be similarly labeled as Table 1, except say HFrEF.
Author Response
Thank you very much for your constructive comments.
1. Editing was done and all changes are highlighted in red font
2. It was modified to “ the endothelium is a single layer of squamous epithelial cells called endothelial cells, which line the inside of all blood vessels”
3. Done
4. The percentage referred to, is the unit for FMD, explained in last paragraph of assessment of EF section (page 4) “the change in brachial artery diameter can be quantified as a percentage change from baseline diameter” (this is in response to reactive hyperemia). As an example, if baseline diameter was 6 mm, and it increased to 6.3 mm, then FMD is 0.3/6 *100=5%
5. Table 2 title has been corrected

Reviewer 3 Report
The review is very interesting. Overall the topic of the review is of relevance for the scientific community and I think worth being published. However, I have some concerns:
The first relevant large-animal model (pig) to study HFpEF has been described in 2015 (Am J Physiol Heart Circ Physiol. 2015;309:H1407-18) and should be mentioned by the Authors.
The following reports on the importance of physical exercise in ameliorating cardiovascular health in HF should be briefly discussed:
-Franks PW. J Am Heart Assoc. 2016 Oct 31;5(11).
-Iaccarino G. Front Physiol. 2013 Aug 12;4:209.
The importance of microRNA in the regulation of endothelial (dye)function (J Cell Physiol. 2016;231:1638-1644) should be addressed.
The functional role of the adrenergic system in linking physical activity and endothelial function in heart failure should be extensively discussed.
The Authors should incorporate a pictorial or cartoon representation of the topics discussed in the Review in order to facilitate the comprehension and increase the overall impact of the manuscript.
Author Response
Thank you very much for your constructive comments.
1. I have added a section under the title role of pharmacological agents on ED in HFpEF, where that study was referred to.
2. The study by Frank et al. was referred to in the beginning of role of exercise training on ED in HFrEF section. While the second study by Iaccarino et al. was discussed in the section of functional role of adrenergic system, physical exercise, and ED in HFrEF patients
3. microRNA implication in endothelial function in CHF patients was briefly mentioned as one of the suggested pathophysiological mechanism of ED in CHF.
4. A new section on “Adrenergic system, physical exercise, and ED in HFrEF patients” has been added. It is a very broad topic but I tried to summarize it to link them together
5. Figure 1 has been added on “ Mechanisms of beneficial effects of physical exercise in HFrEF patients and animal models”
Round 2
Reviewer 3 Report
The Authors did not respond to this concern (despite stating so in the response-to-reviewers):
The first relevant large-animal model (pig) to study HFpEF has been described in 2015 (Am J Physiol Heart Circ Physiol. 2015;309:H1407-18) and should be mentioned by the Authors.
The following reports on the importance of physical exercise in ameliorating cardiovascular health in HF should be briefly discussed:
-Franks PW. J Am Heart Assoc. 2016 Oct 31;5(11).
-Iaccarino G. Front Physiol. 2013 Aug 12;4:209.
Author Response
Thank you very much. Kindly see the attachment.

Round 3
Reviewer 3 Report
The Authors addressed the Reviewers'concerns
Author Response
Thank you very much